# Enhancing the Surface Hydrophilicity of an Aluminum Alloy Using Two-Step Anodizing and the Effect on Inkjet Printing Characteristics

Youngyoon Kim  and Wook-Bae Kim *

Department of Mechanical Design Engineering, Tech University of Korea, Siheung 15073, Republic of Korea
* Correspondence: wkim@tukorea.ac.kr; Tel.: +82-31-8041-0430

**Abstract:** Aluminum alloy anodizing is widely used in the surface treatment industry to provide surface protection and decoration. The resulting anodic aluminum oxide film enables durable printing and dyeing of metals owing to its nanoporous structure, which easily absorbs ink. Conventional one-step anodizing of Al 1050 using sulfuric acid was observed to form a surface with small pore diameters less than 10 nm and lead to an average contact angle of 30°, whereas two-step anodizing yielded a regular pore pattern with significantly larger pores, reducing the contact angle to less than 20°. This change in pore structure and the corresponding enhanced hydrophilicity directly impacted inkjet printing characteristics; inkjet printing of 13 pL droplets on the one-step and two-step anodized surfaces showed that the average dot diameter varied from 72.2 μm to 48.0 μm according to applied voltage and anodizing time. The ink dot diameters on the two-step film were smaller than those on the one-step film produced under the same conditions, and the dot diameters decreased as the average pore diameter increased under an increasing anodizing voltage up to 20 V, indicating improved hydrophilicity. The pore volumes produced by two-step anodizing were larger, facilitating ink droplet absorption during spreading, which was examined by elemental analysis of cross-sections of the ink-filled porous specimen.

**Keywords:** absorption; aluminum alloy; anodizing; inkjet printing; porous structure; wettability



## 1. Introduction

Aluminum alloys are highly recyclable, lightweight metals that exhibit excellent processability and conductivity. Owing to these favorable properties, they are widely used in applications ranging from automobile and aircraft parts to the exterior components of electronic devices. While aluminum alloys are more expensive than steel, they have received increasing attention owing to their excellent specific strength and superior aesthetic effect resulting from their desirable appearance, achieved through various surface treatments. Among such surface treatments, anodizing, which deposits a porous anodized aluminum oxide (AAO) film with a thickness of several tens of μm on the surface of the aluminum, is essential in aluminum product manufacturing. This AAO film imparts corrosion resistance, wear resistance, and thermal control properties to the aluminum component. Furthermore, the AAO coating can prevent color loss from a leaching agent or during handling and use by depositing a coloring agent such as a dye inside the film pores during the anodization process and subsequently applying a sealing treatment [1,2].

As Color, Material, and Finish (CMF) design has become increasingly important, there is a growing need for high-resolution digital printing technology for aluminum [3–6]. The inkjet printing technique selectively deposits colorants by dropping small ink droplets onto a substrate and can play a critical role in bespoke manufacturing. In addition to conventional graphic printing, inkjet printing represents a versatile tool that has become widely used in the digital fabrication of electric circuits for organic thin-film transistors, light-emitting diodes, supercapacitors, fuel cells, and battery electrodes [7–10]. Various conduc-

tors (silver, copper, gold, nickel, etc.), ceramics (silica, alumina, zirconia, titania, etc.), semiconductors (silicon, molybdenum disulfide, graphene), mineral (lithium- and manganese-rich compounds) nanomaterials dispersed in water, organic solvents, ionic liquids, and UV resins have been used as ink materials [11–15].

The quality of inkjet printing largely depends on the droplet ejection properties and the wettability of the ink on the solid substrate. The wettability at the interface where the liquid ink contacts with the porous AAO film is an important phenomenon in inkjet printing as it affects the spreading of the droplets, absorption of the liquid droplets into the surface, and evaporation of the droplets. Since the pore characteristics such as pore diameter, interpore distance, and pore depth (film thickness) of an AAO film are controlled by the applied anodizing conditions such as the electrolyte, voltage, and temperature, it is crucial to understand the relationship between the surface structure characteristics and wettability corresponding to various anodizing and inkjet printing properties.

It has been reported that the porous AAO film surface became hydrophilic to hydrophobic by increasing the size of pores from tens of nm to hundreds of nm. Specifically, the water contact angle changed from 12° to 130° for the pore diameter in the range from about 10 to 400 nm in its original surface chemistry [16–18]. The observed wettability behavior was described as a transition from the Wenzel to Cassie–Baxter state. The pore walls were partially wetted (filled) by water with air trapped within the pores in Wenzel state, and water droplet remained on the top surface and air pockets were trapped underneath the water, revealing that the wetting state is the Cassie–Baxter state. However, in the case of an aluminum alloy such as Al 6061, hydrophilicity has been observed with increasing AAO film thickness even when the average pore diameter is quite large [19]. It was also observed that the evaporation rate was significantly influenced by the pore structure [20].

Various studies have been undertaken to obtain the functional wettability of the aluminum surface via anodizing and a large amount of research has involved the superhydrophobic and/or superydrophilic surface treatment of the aluminum surface [21–29]. A limitation with the above approaches, however, is that the conditions used in these studies are not practical for the surface treatment of aluminum alloy, especially for decorative products requiring printing and coloration. For instance, an efficient cooling system to maintain electrolyte temperature as low under high anodizing voltage conditions and long processing time (~several hours) was necessary in addition to expensive high-purity aluminum with a purity of 99.99% or above [30]. Therefore, aluminum alloys and the anodizing conditions need to be explored further to reduce time and cost.

Recently, it was found that the application of a two-step anodizing process, in which the first oxide layer is removed and anodizing is performed again, using the conventional industrial sulfuric acid anodizing method notably improved the hydrophilicity of water on the AAO surface [31]. The porous AAO layers by sulfuric acid are advantageous in the production of colored surface finishes on aluminum alloys because they absorb solution liquid well due to their hydrophilicity in nature. Hydrophilic behavior of liquid ink on the anodized surface has been shown to exert a substantial impact on decorative color yield and quality and is even more important in digital printing where resolution is critical [32,33].

To the best of the authors' knowledge, there has been no analysis reported on the inkjet printing and wettability characteristics of aluminum alloys. In this study, the effects of two-step anodizing conditions, including anodizing time and electrical potential on the nanostructure characteristics, and water contact angle were investigated and compared to the one-step anodizing. We have focused on the range of industrially applicable anodizing conditions, in which the porous AAO films are stably formed based on the typical process condition without defects such as burning, crack, and over-dissolution [34]. In addition, using an inkjet printer, we dropped water-based ink droplets on the porous AAO surface and analyzed the size of the dots according to the anodizing conditions to quantify the spreading and absorption of the ink. Finally, cross-sections of the ink-filled porous AAO layer formed by the one- and two-step anodizing were examined by energy-dispersive

spectroscopy (EDS) mounted on a transmission electron microscope for the quantitative analysis of ink components.

## 2. Experimental Methods

The experimental method to produce AAO film through two-step anodizing is described in Figure 1. The material used in these experiments was Al 1050 with #8 super mirror finish (ASTM A 480) with a surface roughness (root mean square) of 35 nm. An Al 1050 plate with a thickness of 0.6 mm was cut via chemical etching to prepare 20 mm × 20 mm specimens. Prior to anodizing, 5 min of ultrasonic cleaning was performed using acetone to degrease each specimen before they were immersed in a mixed solution of 6 wt.% phosphoric acid (Duksan Pure Chemical Co., Ansan, Republic of Korea) and 1.8 wt.% chromic acid (Daejung Co., Siheung, Republic of Korea) at 60 °C for 30 min to remove the existing oxide film layer. Anodizing was then performed in a 2 M sulfuric acid solution (Duksan Pure Chemical Co.). During this process, the specimen was fixed using an aluminum rack in a double wall beaker with a volume of 1.5 L, and a platinum grid was employed as the counter electrode, located 5 cm away from the specimen. An external refrigerated and heating bath circulator (RW3-1025, Jeio Tech, Daejeon, Republic of Korea) was used to maintain the temperature of the sulfuric acid at 25 ± 1 °C. One-step anodized specimens were prepared by applying 14, 16, 18, 20, or 22 V of constant voltage for 25 min of anodizing, followed by etching in 5 wt.% sulfuric acid at 60 °C for 1 min. Two-step anodized specimens were prepared by first removing the oxide film previously grown by 14–22 V for 25 min on the one-step anodized specimens using a chromic acid–phosphoric acid mixture, then anodizing them under the same conditions as for the one-step process for 5, 25, or 45 min, then etching them in sulfuric acid. Finally, the specimens were thoroughly rinsed with deionized water and dried prior to analysis.

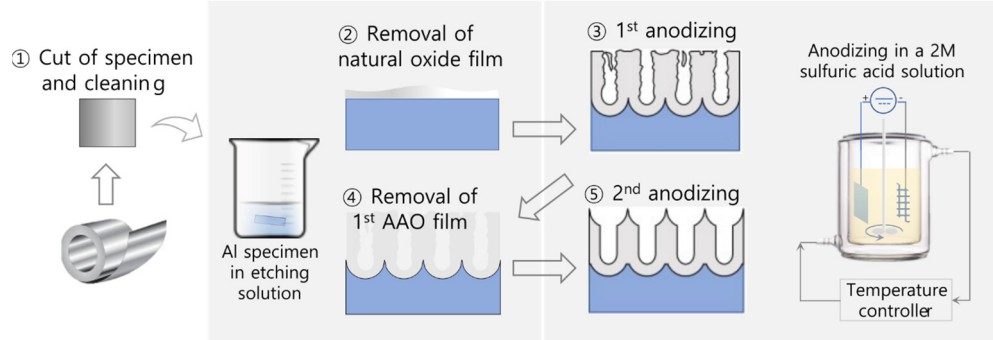

**Figure 1.** Schematic illustration of the two-step anodizing process.

The surface of each specimen substrate was then characterized using scanning electron microscopy (FE-SEM, Nova NanoSEM 450, FEI, Hillsboro, OR, USA). The average pore diameter, porosity, and interpore distance were quantified from the SEM images using ImageJ (version 1.8.0), an open-source Java program. The SEM images were converted into 8 bits to enhance the contrast and then segmented based on thresholding to determine the area of the pore for each sample. Porosity was measured as the fraction of the pores within the entire image layer. On the pore surface, very small (area less than 10 nm$^2$) pores were regarded as defects and excluded from the calculation.

The surface wettability was measured using a contact angle goniometer (L2004A1, Osslia, Sheffield, UK) by ejecting 10 µL of water droplets at a speed of 5 µL/s and collecting images for 3 s. Five measurements were performed per sample.

Printing was performed on each anodized specimen by attaching a printer head (Dimatix DMC 11610, Fujifilm, Tokyo, Japan) to an inkjet printer (Omnijet 300, Unijet, Pyeongtaek, Republic of Korea) and using a nozzle to deposit XL30 ink (Dimatix, Fujifilm), which has a density of 1.082 g/cm$^3$, a viscosity of 10.6 cP, and a surface tension of

38 dynes/cm$^2$. The ambient temperature and relative humidity during printing were 22 °C and 32%, respectively, and the temperature of the substrate was 13 °C.

In piezoelectric printing, ink droplets were discharged by applying voltage as a trapezoidal waveform. Printing conditions are shown in Table 1.

**Table 1.** Inkjet printing conditions.

| Voltage (V) | Distance of Nozzle to Sample (mm) | Frequency (Hz) | Drop Velocity (m/s) | Drop Volume (pL) | Resolution (DPI) |
|---|---|---|---|---|---|
| 40 | 1 | 2000 | 4.90 ± 0.5 | 13 ± 0.05 | 400 |

After printing, each specimen was dried for 60 min in an environment free from external factors such as wind and was subsequently imaged using SEM for surface characterization.

Finally, two anodized samples were dipped into the solvent-based inkjet ink (A2-S, Atech Co., Hanam, Republic of Korea) for 30 s and then stored in an oven to dry at 60 °C for 6 h. The cross-sections of two samples were observed using field-emission scanning electron microscopy (FE-SEM, JSM-6700F, JEOL, Akishima, Japan). The samples to be examined by TEM were thinned using a focused ion beam (FIB, Quanta 3D FEG, FEI, Hillsboro, OR, USA). The structure and composition of the thin cross-section tunnels were characterized using field-emission transmission electron microscopy (TEM, Tecnai G2 F20, FEI, Hillsboro, OR, USA) and energy dispersive X-ray spectroscopy (EDS, FEI, Hillsboro, OR, USA).

## 3. Results and Discussion

### 3.1. Surface Morphologies of AAO Films

Figure 2a presents the SEM images of the porous AAO film surfaces according to the voltage applied during the sulfuric acid-based one-step and two-step anodizing of Al 1050 with constant temperature (25 °C) and time (25 min). The pore size increase with increasing applied voltages similarly in both one- and two-step processes. Additionally, the pore diameter and porosity on the surfaces of the specimens subjected to two-step anodizing are distinctly larger than the conventional one-step anodizing process, except in the 22 V condition. Figure 2b,c show the average pore diameter and porosity, respectively, obtained by analyzing the SEM images of the anodized surfaces using ImageJ, revealing that the average pore diameters of the specimens subjected to two-step anodizing increased by 54–90% compared to those of the specimens subjected to one-step anodizing under the same applied voltage; this in turn led to a 97–192% increase in porosity. In the case of 22 V conditions in the two-step anodizing, the nanoscale vertical pore structure collapses due to excessive dissolution of the AAO layer, resulting in hierarchical micro-nano structure: an uneven microstructure with nanofibrous surface. Figure 1 presents the interpore distance, which is larger by two-step anodizing and the increase with the applied volage is evident. In one-step anodizing, since many defects such as branched pores and pits exist on the surface, a relatively large number of unstructured pores are formed, and the interpore distances change little with voltage.

Figure 3 compares the SEM images of the fractured AAO layer resulting from one-step and two-step anodizing under applied voltages of 16 and 20 V, showing clear differences between the surface roughnesses and cross-sectional channels of the one-step and two-step layers. In particular, the local deformations present on the surface of the raw specimen remained on the one-step anodized surface. The surface resulting from the two-step anodizing process was observed to be relatively flat owing to the removal of the film layer deposited by the previous anodizing step and characterized by a concave hexagonal honeycomb structure pattern with sharp edges. These features are the result of the regular hexagonal pattern that was formed on the base aluminum during the first anodizing

step and subsequently formed on the film surface during the second anodizing step. Furthermore, while the channel diameter was smaller and the pore inlet slightly irregular in the cross-section of the one-step anodized coating, the channels were relatively more consistent and straighter in that of the two-step coating.

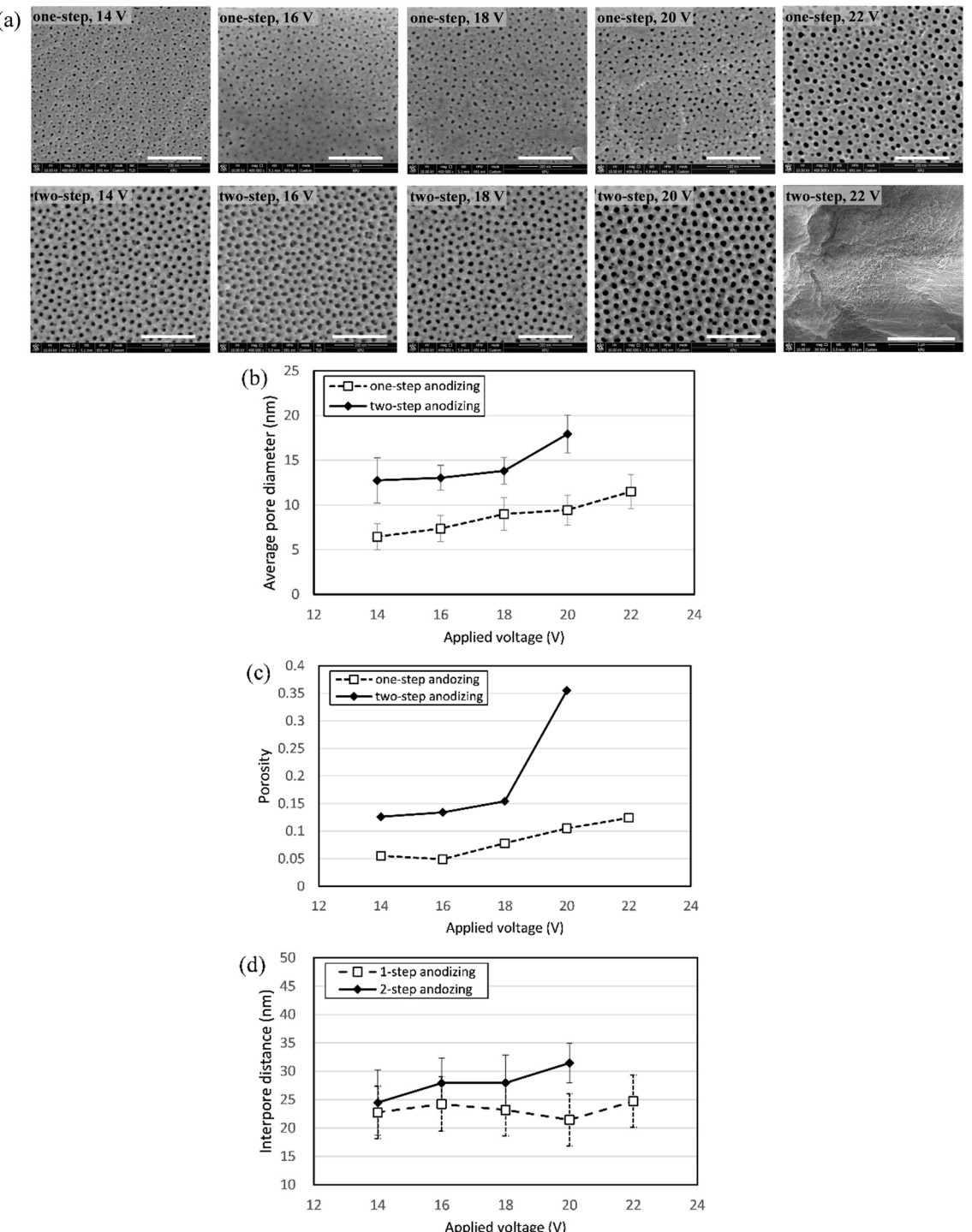

**Figure 2.** (**a**) SEM images of the AAO layer created by one-step and two-step anodizing at static voltages of 14, 16, 18, 20, and 22 V for 25 min (the scale bar is 200 nm for all except 2 μm in the image of two-step, 22 V), (**b**) the corresponding change in average pore diameter, (**c**) the corresponding change in porosity, and (**d**) the corresponding change in interpore distance.

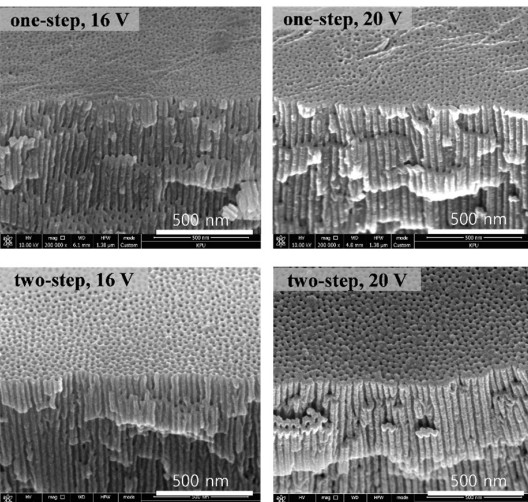

**Figure 3.** Top and cross-section views of the AAO films produced by one-step and two-step anodizing at 16 and 20 V.

Figure 4a shows the AAO film surfaces resulting from one-step and two-step anodizing at 20 V for 5 min and 45 min. Compared with the images of the surfaces after 25 min of anodizing in Figure 2a, it can be observed that both the pore diameters and porosity increased with increasing anodizing time. After 5 min of anodizing, as shown in Figure 4b,c, there was almost no difference between the pore diameters and porosities produced by one- and two-step anodizing. However, the initial regular hexagonal pattern in the pore structure was more evident after two-step anodizing owing to the more uniform formation of pores with increasing anodizing time; this eventually led to an increase in the pore diameters and porosity, suggesting that the pore growth rate during two-step anodizing is higher than that during one-step anodizing. Figure 3 shows that the interpore distance by two-step anodizing is constant with time and larger than that by one-step anodizing. This is well explained by the mechanism of two-step anodizing, in which the uniform dimples formed on the aluminum surface by the first anodizing act as initiation sites for growing an ordered and straight porous structure [35,36]. Film thickness was proportional to time and almost the same in both as demonstrated in Figure 4e. From the results of Figure 4, it is known that the thickness of the AAO film increases equally with time, but the pore diameter and porosity are higher in two-step anodizing due to the regular arrangement of pores and fewer defects than one-step anodizing.

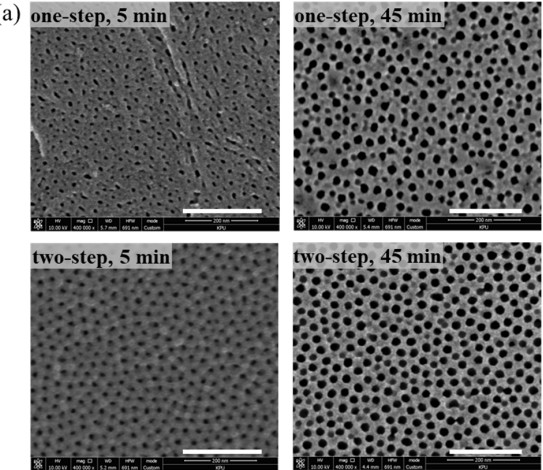

**Figure 4.** *Cont.*

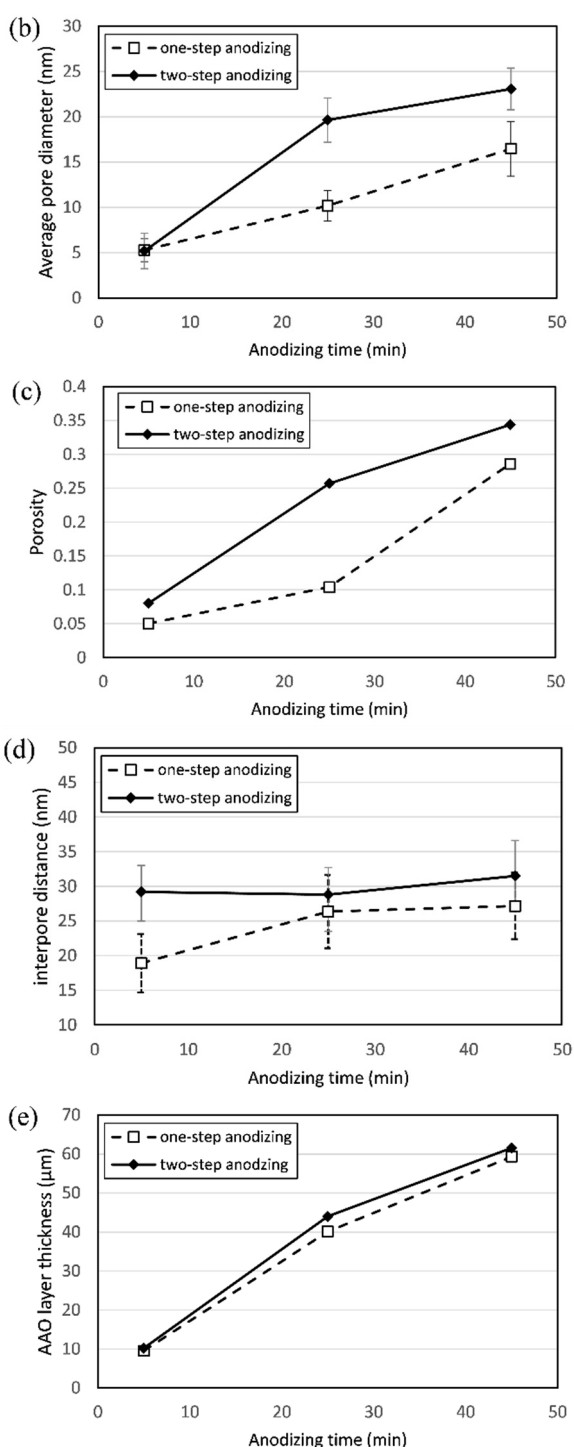

**Figure 4.** (**a**) SEM images of the top of the AAO film produced by one-step and two-step anodizing at an applied voltage of 20 V for 5 and 45 min (scale bar 200 nm), and the (**b**) average pore diameter, (**c**) porosity, (**d**) interpore distance and (**e**) is film thickness after 5, 25, and 45 min of anodizing.

### 3.2. Wettability Characteristics

Figure 5 presents the static water contact angles on the anodized surfaces shown in Figure 2. When anodizing was conducted under an applied voltage of 14 V, the contact angles with the AAO film produced by one-step and two-step anodizing were at 44.3° and 29.2°, respectively, which was affected by different pore size and porosity. In the 16–20 V range, the contact angle decreases linearly with voltage under both anodizing conditions and the amount of this decrease was much more substantial on the surface produced by

two-step anodizing, for which the contact angle decreased as low as 16.2° at 20 V. The contact angle decreases more rapidly at 22 V to enhance the hydrophilicity, and the AAO surface in the two-step anodizing became superhydrophilic as the contact angle was close to zero due to the hierarchical nanofibrous structure, caused by over-erosion of the nanopore structure.

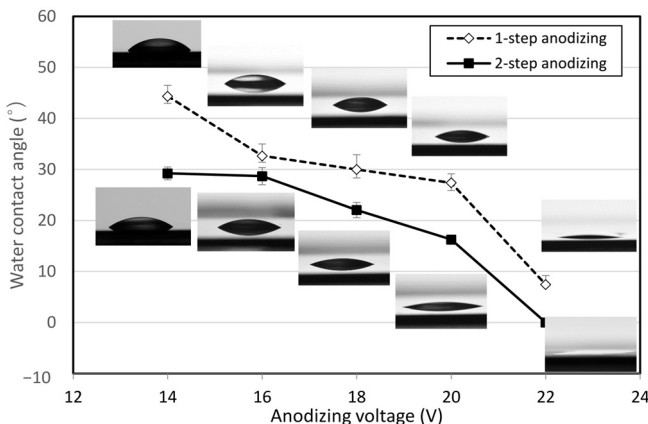

**Figure 5.** Static water contact angle on the AAO films produced by one-step and two-step anodizing with applied voltages of 14–22 V for 25 min.

The pore structure the AAO film described by the pore diameters and depths can be tuned by adjusting the anodizing electrolyte, voltage, and processing time, changing the surface roughness and directly affecting the wettability of the surface. This effect can be explained by the Wenzel and Cassie–Baxter models [37,38], respectively, given by:

$$cos\theta_W = rcos\theta_S \tag{1}$$

$$cos\theta_{CB} = f(1 + cos\theta_S) - 1 \tag{2}$$

where $\theta_S$ is the equilibrium contact angle on a smooth surface in Young's equation, $\theta_W$ and $\theta_{CB}$ are the respective apparent contact angles in the Wenzel and Cassie states, $r$ is the surface roughness factor (defined as the ratio of the actual area to the projected area of the surface), and $f$ is the solid fraction in the solid–liquid interface at the base of the droplet.

A smooth aluminum oxide film that has not been anodized exhibits a contact angle of 62–85° and thus is chemically hydrophilic [16,17,39,40]. In the pore structure of high-purity aluminum subjected to two-step anodizing, the AAO film will gradually change from hydrophilic to hydrophobic as the pore diameters increase. The contact angle is generally proportional to the average pore diameter, but when the pore diameters are greater than 100–200 nm, air trapped in the pores supports the water droplets and prevents them from penetrating into the pores, resulting in wettability characteristics reflecting the Cassie state. On the other hand, when pore diameters of 5–30 nm are formed by sulfuric acid-based anodizing, the droplets are in the Wenzel state and penetrate into the pores by capillary force, thereby sufficiently wetting the contact surface. Indeed, hexagonal dimples were clearly formed at the pore entrances on the two-step anodized specimens, and the corresponding surface roughnesses were greater than those of the corresponding one-step anodized specimens. In other words, the roughness factor was larger on the two-step anodized surfaces, further improving their hydrophilicity.

The amount of water penetrating into the pore channel may also be a factor affecting the contact angle. Figure 6 depicts a liquid droplet placed on a pore in an anodized aluminum surface and its penetration into the channels via capillary force, where air trapped in the channels resists complete wetting. At this point, assuming that the trapped

air is an ideal gas and that the product of pressure and volume is invariant, the following relationship can be obtained:

$$P_g - P_a = P_a \frac{h_1}{H - h_1} \tag{3}$$

where $P_g$ is the pressure of the trapped air, $P_a$ is the atmospheric pressure, $H$ is the depth of the pore channel, and $h_1$ is the droplet penetration depth. Given the low height of the liquid droplet, the effects of hydrostatic pressure and gravity inside the droplet were considered negligible. The pressure equilibrium equation then becomes:

$$(P_a - P_g)\frac{\pi D_p^2}{4} + \gamma cos\theta \ \pi D_p = 0 \tag{4}$$

where $D_p$ is the diameter of the pore and $\gamma$ is the surface tension of water (72 mN/m). Rearranging Equations (3) and (4) then leads to the following expression for the penetration ratio:

$$\frac{h_1}{H} = \frac{4\gamma cos\theta}{4\gamma cos\theta + P_a D_p} \tag{5}$$

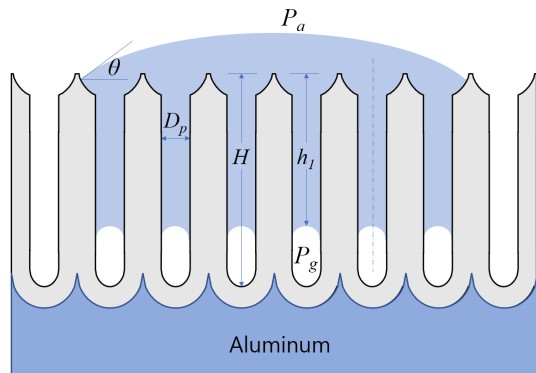

**Figure 6.** Liquid droplet on the nanopore structures produced by one-step and two-step anodizing.

Owing to the small average pore diameter and low contact angle, the penetration ratio given by Equation (5) generally becomes $\frac{h_1}{H} > 0.99$ for the pore layers produced under one-step and two-step anodizing conditions.

Figure 7 shows the change in the water droplet contact angle on the AAO film produced by one-step and two-step anodizing for 5, 25, and 45 min while applying a constant 20 V. On the one-step anodized surface, the contact angle decreased from 30.6° to 20.2° as the processing time increased from 5 to 45 min, primarily owing to the increase in surface pore inlet diameter and channel depth over time. In contrast, the contact angle on the two-step anodized surface was much smaller even though the average pore diameter and porosity were similar to those on the one-step anodized surface after the initial 5 min of anodizing. This result is considered to be related to the increased surface roughness induced by the hexagonal dimple structure generated by the two-step anodizing after the removal of the AAO film produced by the first anodizing step. In fact, despite the distinct increase in average pore diameter over time for two-step anodizing as shown in Figure 4, the change in contact angle with anodizing time was less pronounced compared to that for the one-step anodizing.

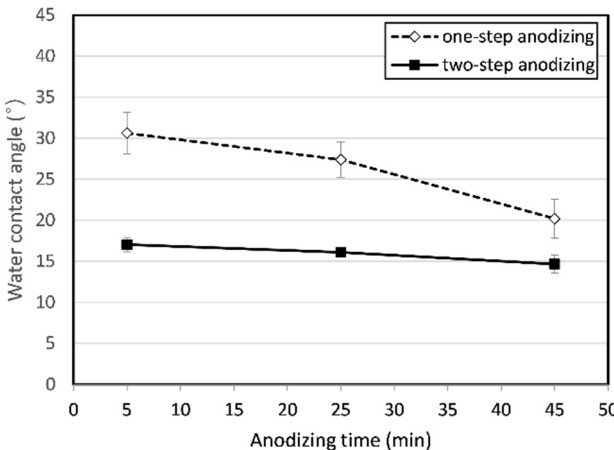

**Figure 7.** Static water contact angle on the AAO films produced by one-step and two-step anodizing at an applied voltage of 20 V for 5 min, 25 min, and 45 min.

### 3.3. Inkjet Printing Characteristics: Dot Size and Absorption into Porous AAO Layer

Figure 8 shows micrograph images of the dot pattern produced when inkjet printing with one drop volume of 13 pL per dot on the one-step and two-step AAO films as well as measurements of the dot diameters. Anodizing was performed at three applied voltages, 16, 18, and 20 V, to prepare a nanoporous film absorbing liquid ink. The dot size clearly decreased as the applied voltage increased for all one-step and two-step AAO films. The dark spots formed at the aluminum interface under the oxide layer gradually faded with increasing anodizing voltage as the thickness of the film increased. The dot size measured from the image analysis indicated that the dot diameter was smaller on the two-step AAO film than on the one-step AAO film produced under the same applied voltage. In addition, while the standard deviation of the pore diameters was large at 16 V, it decreased as the anodizing voltage increased.

Inkjet droplets are much smaller in volume—on the picoliter scale—compared to the water droplets used for contact angle measurements. Therefore, factors relevant to the size of the inkjet dots on the nanopore surface include the contact angle as well as absorption into the pore channel and evaporation. However, in the case of a water droplet with a volume of tens of picoliters, the evaporation time scale is of the order of $10^{-1}$ s, while spreading and absorption occur on relatively shorter time scales. Thus, the effect of evaporation on inkjet dot size was expected to be minimal [32,41,42].

To provide a base case scenario, the experimental diameter of the inkjet dot on the porous AAO surface was compared to the theoretical contact diameter when an inkjet droplet was dropped on a non-porous surface. The contact radius of a droplet placed on a smooth, non-porous surface can be expressed as a function of volume and contact angle as follows:

$$r = \left[ \frac{3V}{\pi} \frac{sin^3\theta}{2 - 3cos\theta + cos^3\theta} \right]^{1/3} \tag{6}$$

where $V$ is the volume of the liquid droplet.

Assuming that the contact angle of the ink droplet is equal to the water contact angle in Figure 5 allows for the use of Equation (6) to determine the contact radius of an inkjet droplet on an AAO film according to the applied voltage for both one-step and two-step anodizing. The results are presented in Figure 9, which shows that the contact diameter on a non-porous surface decreases as the contact angle increases considering the droplet volume of 13 pL; however, the experimentally obtained relationship between inkjet dot diameter and the contact angle of the water droplet on the AAO film indicates that the inkjet dot size decreased with decreasing contact angle. In particular, the contact diameters on the two-step AAO films produced at 18 and 20 V were significantly smaller than those predicted from the contact angle on a non-porous surface.

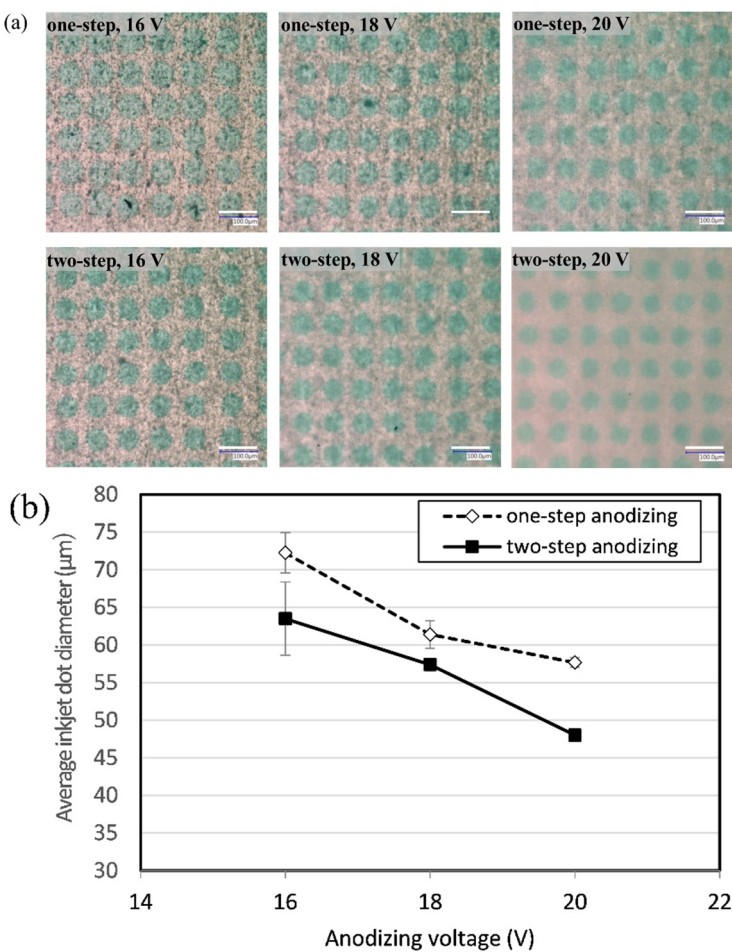

**Figure 8.** (**a**) Images of inkjet dot patterns on the AAO films produced by one-step and two-step anodizing with applied voltages of 16, 18, and 20 V (scale bar 100 µm), and (**b**) the change in dot diameter (a drop volume of 13 pL) according to applied anodizing voltage.

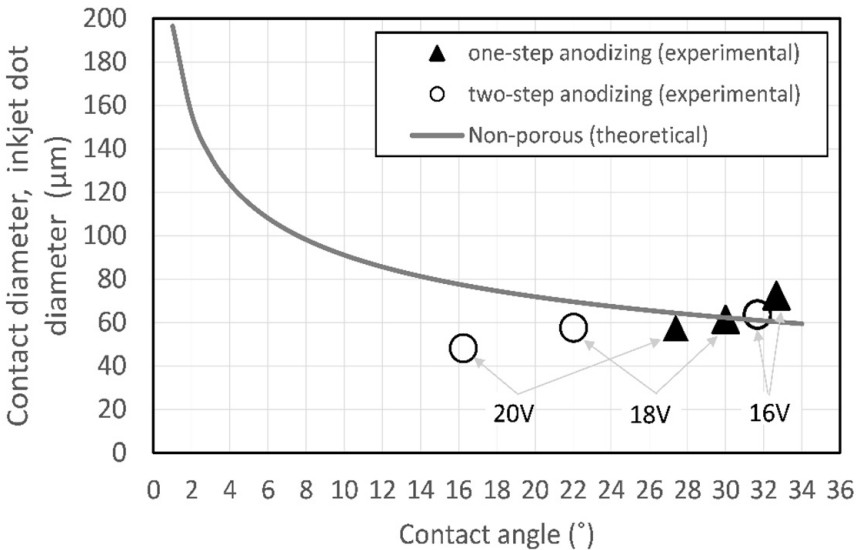

**Figure 9.** Theoretical droplet contact diameter according to contact angle on a non-porous surface and experimental inkjet dot diameters according to the water contact angles on different AAO films.

The pore volumes capable of absorbing the ink droplets can be calculated from the porosity and AAO film thickness by measuring the diameters of the inkjet dots printed on the specimens' surfaces obtained under the different experimental conditions evaluated in this study. That is, assuming a full wetting state in which the liquid is completely absorbed into the pores in contact with the droplets, the maximum quantity of absorbable ink can be calculated by considering the porosity and film thickness. Figure 10 shows the pore volume underneath a 13 pL inkjet droplet on each AAO film surface. On the one hand, it was estimated that pore volumes of 1.4 and 3.9 pL were formed in the one-step and two-step films, respectively, under a low applied voltage (16 V), such that the entire ink droplets deposited on these surfaces could not be fully absorbed. On the other hand, most of the droplets could be absorbed by the pore layers of the two-step AAO films produced at 18 and 20 V, and by the pore layers of the one-step AAO films produced at 20 V, as these voltage conditions led to the formation of a pore volume corresponding to greater than 95% of the 13 pL ink droplet volume. In this case, it is expected that the ink will rapidly penetrate into the pores and that the spreading of the ink droplet will therefore be limited. In other words, as shown in Figures 7 and 8, it is believed that the ink dot diameters on the two-step AAO films produced at voltages greater than 18 V will be smaller than the value predicted using the non-porous surface because of the increased pore volume and enhanced wettability, which, in turn, lead to the absorption of most of the droplet volume during spreading.

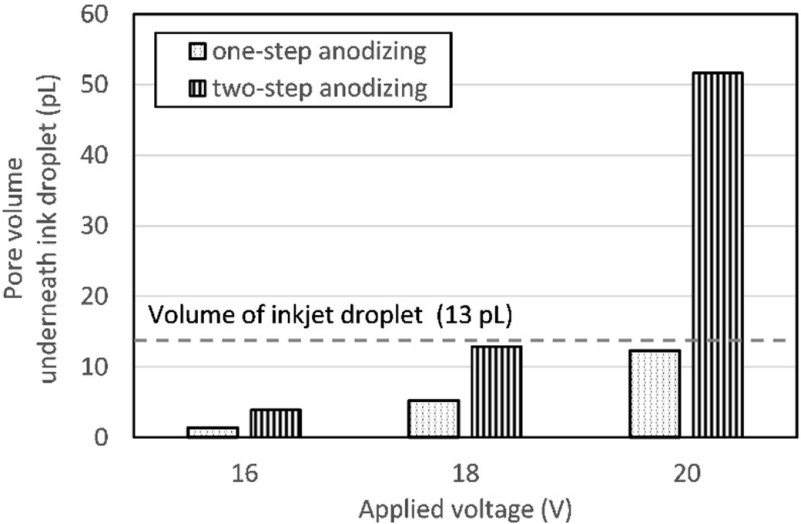

**Figure 10.** Pore volumes beneath a 13 pL inkjet droplet on one-step and two-step AAO films according to applied voltage.

Considering that the volume of a drop in digital inkjet printing is usually 15–20 pL for a 600 dpi × 600 dpi resolution, it is clear that the anodizing condition affects the spreading and absorption behavior of ink on the AAO surface, and thus plays an important role in controlling printing quality.

Figure 11 shows a 0.5 mm × 0.5 mm square patten printed at 450 dpi on the one-step and two-step AAO films according to the applied voltage and presents different printing qualities depending on anodizing conditions. In one-step AAO of 16 V, the square boundary is blurred and the color concentration is not uniform because the substrate cannot absorb the deposited ink, but the boundary becomes clear as the voltage increases. In two-step AAO films, all patterns have clear boundaries, and color shade becomes intense due to the increase in the amount of ink absorbed as the voltage increases. Printing characteristics such as color and resolution can be adjusted by enhancing surface wettability according to changes in the nanoporous structure through two-step anodizing.

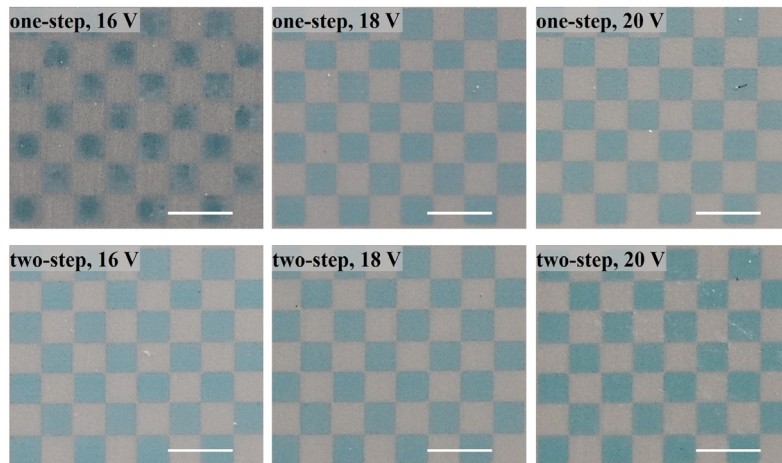

**Figure 11.** Images of inkjet-printed 0.5 mm × 0.5 mm square arrays on the AAO films produced by one-step and two-step anodizing with applied voltages of 16, 18, and 20 V (scale bar 1 mm).

Figure 12 presents the quantitative analysis of the ink absorption into the AAO layers according to the anodizing conditions using solvent-based inkjet ink applied to anodized aluminum industrially. The one-step and two-step anodizing samples by the anodizing condition of 20 min at 16 V and 20 V, respectively, were dipped in solvent-based ink and sufficiently dried. Figure 12a,b show a SEM image of the electron transparent thinned foil cross-section window, as prepared using the FIB technique for the one-step and two-step anodized layer, respectively. Also included are TEM images near the top and in the middle (4 µm below the top), locations indicated by squares in the SEM image. After the ink dipping and drying process, it may be seen that there is no residue on the two-step anodized surface at 20 V, while the dye residue remains thick on the one-step anodized layer at 16 V, on which the ink is less wetted. Figure 11 shows the bright-field TEM image and the corresponding EDS map for the top AAO layer with dye residues of the one-step AAO film, square positions @in Figure 12a. The deposited dye residue/AAO interface can be distinguished from TEM and EDS images. It is obvious that the dye layer is mostly composed of carbon, which is a component that usually makes up about 60% of the ink but is not present in the AAO layer. Additionally, inorganic additives such as silicon and sulfur were also detected in the ink layer. Figure 11 compares the element composition at each point indicated by the square in Figure 12a, obtained from the EDS analysis. At the interface, the carbon content occupies a high ratio of 35.5% due to the dye residue. Inside the AAO nanopores, higher contents of ink components such as carbon, silicon, and fluorine were detected in the two-step AAO layer than in the one-step layer. In both AAO layers, the upper part of the AAO layer has a higher carbon content than the middle part. From this elemental analysis, it can be seen that the high wettability of the two-step AAO layer increases the ink absorption and decreases the ink penetration with the nanopore depth.

(a)

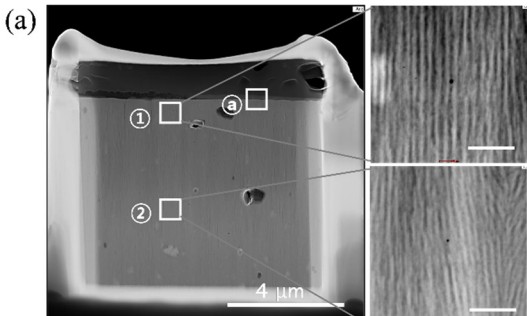

**Figure 12.** *Cont.*

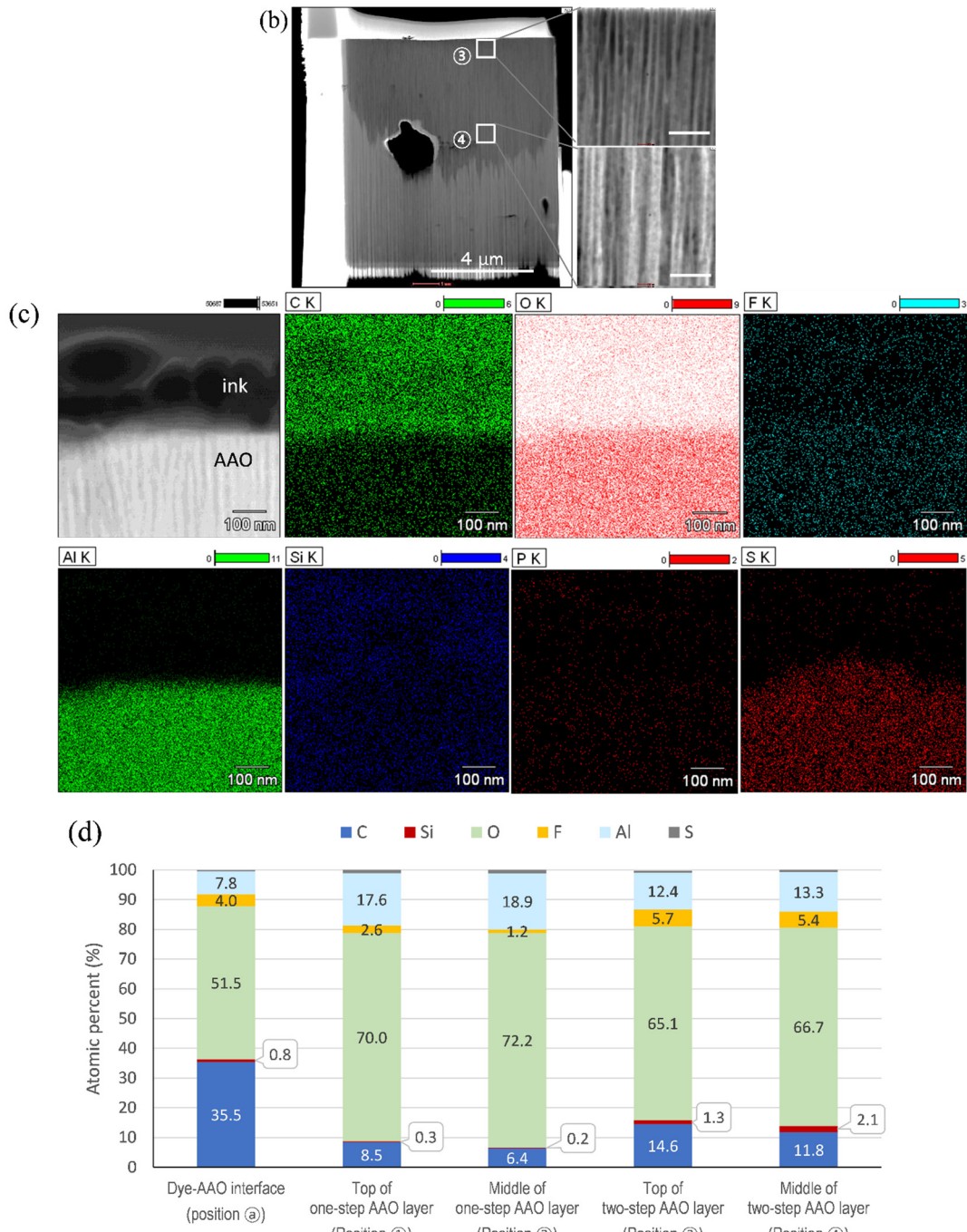

**Figure 12.** TEM-EDS measurement results for the one-step and two-step AAO layers wetted in solvent-based ink and dried. (**a**,**b**) SEM images of thinned foil cross-section and TEM images (scale bar 200 nm) of near top surface and middle positions in the oxide film for the one-step and two-step AAO layer, (**c**) EDS elemental maps of the interface with the deposited dye residue on the one-step AAO layer (ⓐ in (**a**)), and (**d**) comparison of elements composition at different positions (ⓐ, ①–④).

## 4. Conclusions

This study demonstrated that sulfuric acid-based two-step anodizing of Al 1050 material can improve the wettability of the one-step anodized surface and control factors related to inkjet printing quality.

Under the same conditions of electric field and time, the AAO film growth rates of the one-step and two-step anodizing were quiet similar. However, the two-step anodizing increased the pore diameter and porosity of the AAO layer more rapidly than the one-step

anodizing and produced straight pore channels with larger entrance diameters. In addition, hexagonal corners were sharply formed around the entrances of the pores, leading to an increase in surface roughness. These effects enhance the hydrophilicity of the two-step anodized surface, reducing the water contact angle compared to the one-step anodized surface.

Under an applied voltage of 20 V, increasing the anodizing time from 5 min to 45 min resulted in a decrease in contact angle from 30.6° to 20.2° for the one-step anodized film, while the contact angle for the two-step anodized film was 17.0° at 5 min and only slightly decreased to 14.0° as the anodizing time increased to 45 min. That is, the application of two-step anodizing alone is considered sufficient to enhance hydrophilicity even under short anodizing times.

Printing an array of water-based ink droplets and measuring the resulting dot sizes indicated that the dot diameter decreased with increasing average pore diameter in the AAO film. In particular, the dot diameter on the two-step anodized surface was small because the ink liquid was easily absorbed into the pores during droplet spreading owing to the expanded internal pore volume in the AAO film with enhanced hydrophilicity. As such, the decrease in the diameters of the inkjet dots and their corresponding low standard deviations indicate an increased achievable inkjet printing resolution.

In addition, it was found through TEM-EDS analysis that the ink components penetrated more and deeper into the pores of the two-step anodizing specimen compared to the one-step specimen when both specimens were immersed in the solvent-based ink. The high absorbency of the two-step anodized film allows for an enhanced durability of the printing surface in external environments.

Therefore, the subsequent printing quality can be well controlled by simply performing anodizing once more with the same conditions in the aluminum anodizing process. More importantly, since wettability can be controlled by changing nanoscale surface morphology and pore channel structure by combining a wide range of process conditions of anodization, two-step anodizing can be applied not only to printing but also to various processes such as coating, drying, and bonding.

**Author Contributions:** Conceptualization, W.-B.K.; Methodology, Y.K. and W.-B.K.; Validation, W.-B.K.; Formal analysis, W.-B.K.; Investigation, Y.K.; Resources, Y.K. Writing—original draft, Y.K.; Writing—review & editing, W.-B.K.; Visualization, W.-B.K.; Project administration, W.-B.K. All authors have read and agreed to the published version of the manuscript.

**Funding:** This work was partly supported by the GRRC program of Gyeonggi Province ((GRRC-KPU2020-B01), High speed additive manufacturing of high strength, high heat-resistant, heat conductive parts for automotive industry) and the Individual Researcher Program (2018R1D1A1B07050525) through the National Research Foundation of Korea (NRF) funded by the Ministry of Education.

**Institutional Review Board Statement:** Not applicable.

**Informed Consent Statement:** Not applicable.

**Data Availability Statement:** Not applicable.

**Conflicts of Interest:** The authors declare no conflict of interest.

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
