# Peer review of "Enhancing the Surface Hydrophilicity of an Aluminum Alloy Using Two-Step Anodizing and the Effect on Inkjet Printing Characteristics"

_coatings, doi:10.3390/coatings13020232_

Round 1

Reviewer 1 Report

The paper is technically sound and the way to calculate and approach the data interpretation and modelling is logical and can improve the understanding of findings of authors. The text is well written and easy to follow, images are mostly clear and self-explaining and the illustrations are easing the understanding of the problem and solution.

I have some questions for authors:

In fig. 1.  you are giving the results for porosity, it would be good to mention the way how you calculate porosity, I suppose it is from the image of the surface but you should precise this in paragraph preceding the image.

From the SEM images in fig. 1a the surface of two step anodizing seam to have pores that are more homogeneously spaced. It seams that at once anodized surfaces there are spaces that contain no pores at all. So, I suppose that you can calculate the average distance of pore centers using some triangulation approach and see if this observation stands. This would give one more argument for the discussion and the need to use the two step process.

What is the reason why the pores are larger when the anodizing lasts longer. It is obvious from images that it is the trend but it would be good to explain this. From the mechanism of the process you propose in your introduction, the first step of anodizing is creating small pits on the surface and when this layer is removed than in the second step those pits are serving as the "nest" for the growth of the final layer. If this is so than the pores should be influenced by this step. From the cross section view it is clear that the pores are tube like and that their cross section seam to be the same in the all layer. Probably that with time the walls are getting thinner and the pores are getting more volume. This could also be explained by measuring the average distances from pore centers for different anodizing times. This would prove the mechanism of the surface development. If you find the distances of pore centers to be similar than the pores are not cylindrical they are in form of a cone that is having larger diameter as the process is progressing.

In the measurements of contact angle authors are giving the results only and we don't have the opportunity to see the experimental measurement of the contact angle. One such image would be welcome in methodology description in the paper.

I liked the analysis of the ink jet printing on the surface, it would be good to give the elemental analysis of the ink in one line and to compare it to the results of elemental analysis performed on the surface of the material.

Author Response

We appreciate your valuable suggestions regarding the formation process of anodic nanostructure. We have made every effort to incorporate your feedback into our revised manuscript. Please refer to the attached pdf file for our response. 

Reviewer 2 Report

This manuscript, entitled „ Enhancing the surface hydrophilicity of an aluminum alloy using two-step anodizing and the effect on inkjet printing characteristics” is relevant to the scope of this journal.

It is a great subject that can offer important knowledge to industry experts.

The authors made a good synthesis of the literature that provides an overview of the research evolution in this area.

Therefore, the article can be recommended for publication only after mandatory revision according to the following suggestions:

1.  The authors mention that "Recently, the present authors found that the application of a two-step anodizing in which the first oxide layer is removed and anodizing is performed again, using the conventional industrial sulfuric acid anodizing method [23]". I think the references should be checked. Reference 23 is not recent and does not seem to be by the same authors...

2. The authors must specify how the surface roughness was established for the samples that were to be anodized.

3. The characteristic parameters in the case of analyses using the SEM microscope must be specified.

4. Why didn't the authors choose to use the same scale in the images in Figure 2?

5. The first image in Figure 3a should be replaced with a clearer one.

Author Response

Thank you for your comments. We were able to correct errors in the paper and improve its quality. We have made every effort to incorporate your feedback as much as possible. Please refer to the attached pdf file.

Reviewer 3 Report

Dear Authors,

Please make the following changes.

1. It is necessary to reduce the abstract to 200 words.

2. Keywords must be arranged in alphabetical order

3. In the introduction, need to add the types of ink and their chemical composition that can be printed with this method on this template, for example,

https://doi.org/10.1016/j.ceramint.2020.06.318

https://doi.org/10.1021/acsami.0c11846

https://doi.org/10.1002/ente.201901086

4. In the experimental methods, manufacturers of materials, chemicals, and equipment must be indicated.

5. It is necessary to add the ink preparation point and the study of rheological characteristics (Density , Dynamic viscosity, Surface tension of ink, Z number) to be able to print. Specifying the parameters used in piezoelectric printing (voltage, the distance between the nozzle and the sample, etc.) is also necessary.

Author Response

We appreciate very much your comments for improving our paper.

We have made revision to the manuscript based on your comment and explain the details in the comments below.

Round 2

Reviewer 1 Report

Dear Authors,

I was glad seeing that you incorporated some of my comments and you answered the questions well.

I also see that you made some small corrections that I didn't ask for and I appreciate that.

The paper is now ready to be published.

Reviewer 3 Report

Good job